# Chagas cardiomyopathy in Boston, Massachusetts: Identifying disease and improving management after community and hospital-based screening

Katherine A. Reifler[1]*, Alyse Wheelock[2], Samantha M. Hall[3], Alejandra Salazar[1], Shahzad Hassan[4], John A. Bostrom[5], Elizabeth D. Barnett[6], Malwina Carrion[7], Natasha S. Hochberg[1], Davidson H. Hamer[1,8,9], Deepa M. Gopal[5ᐤ], Daniel Bourque[1ᐤ]

1 Section of Infectious Disease, Department of Medicine, Boston University Chobanian & Avedisian School of Medicine, Boston, Massachusetts, United States of America, 2 Section of Preventative Medicine and Epidemiology, Boston University Chobanian & Avedisian School of Medicine, Boston, Massachusetts, United States of America, 3 Department of Environmental Health, Boston University School of Public Health, Boston, Massachusetts, United States of America, 4 Department of Internal Medicine, Boston University Medical Center, Boston, Massachusetts, United States of America, 5 Cardiovascular Division, Department of Medicine, Boston University Chobanian & Avedisian School of Medicine, Boston, Massachusetts, United States of America, 6 Section of Pediatric Infectious Disease, Department of Pediatrics, Boston Medical Center and Boston University Chobanian & Avedisian School of Medicine, Boston, Massachusetts, United States of America, 7 Boston University College of Health and Rehabilitation Sciences: Sargent College, Massachusetts, United States of America, 8 Department of Global Health, Boston University School of Public Health, Boston, Massachusetts, United States of America, 9 Center for Emerging Infectious Disease Policy & Research, Boston University, Boston, Massachusetts, United States of America

ᐤ These authors contributed equally to this work.
* Katherine.reifler@bmc.org

## Abstract

### Background

Limited data exist regarding cardiac manifestations of Chagas disease in migrants living in non-endemic regions.

### Methods

A retrospective cohort analysis of 109 patients with Chagas disease seen at Boston Medical Center (BMC) between January 2016 and January 2023 was performed. Patients were identified by screening and testing migrants from endemic regions at a community health center and BMC. Demographic, laboratory, and cardiac evaluation data were collected.

### Results

Mean age of the 109 patients was 43 years (range 19–76); 61% were female. 79% (86/109) were diagnosed with Chagas disease via screening and 21% (23/109) were tested given symptoms or electrocardiogram abnormalities. Common symptoms included palpitations (25%, 27/109) and chest pain (17%, 18/109); 52% (57/109) were asymptomatic. Right bundle branch block (19%, 19/102), T-wave changes (18%, 18/102), and left anterior fascicular

**Data Availability Statement:** All relevant data are within the manuscript and will be within its Supporting Information files.

**Funding:** KAR's salary is supported by the National Institute of Allergy and Infectious Diseases training grant, T32 5T32AI052074-17. AW's salary is supported by the National Heart, Lung, and Blood Institute training grant, T32HL125232-07. DB and DHH each receive salary support from the Center for Disease Control and Prevention cooperative agreement, NU2GGH002322-01-00. The funders had no role in study design, data collection and analysis, decision to publish, or preparation of the manuscript.

**Competing interests:** I have read the journal's policy and the authors of this manuscript have the following competing interests: After completing this study, NSH was employed by Novartis Institutes for BioMedical Research for which she receives a salary and stocks. NSH, DB, and DHH have collaborated with Kephera Diagnostics on the National Institutes of Health (NIH) Small Business Innovation Research (SBIR) grant funding research for Chagas disease diagnostics and biomarkers

block (11%, 11/102) were the most common electrocardiogram abnormalities; 51% (52/102) had normal electrocardiograms. Cardiomyopathy stage was ascertained in 94 of 109 patients: 51% (48/94) were indeterminate stage A and 49% (46/94) had cardiac structural disease (stages B1-D). Clinical findings that required clinical intervention or change in management were found in 23% (25/109), and included cardiomyopathy, apical hypokinesis/aneurysm, stroke, atrial or ventricular arrhythmias, and apical thrombus.

## Conclusions

These data show high rates of cardiac complications in a cohort of migrants living with Chagas disease in a non-endemic setting. We demonstrate that Chagas disease diagnosis prompts cardiac evaluation which often identifies actionable cardiac disease and provides opportunities for prevention and treatment.

### Author summary

Chagas cardiomyopathy is a manifestation of end-organ damage from *Trypanosoma cruzi* parasitic infection. Most infections occur in Mexico, Central and South America but disease is present globally due to migration. Approximately 30% of those with Chagas disease develop cardiomyopathy years after infection from persistent heart muscle and conduction system inflammation, which can be life-threatening. Unfortunately, Chagas disease remains underdiagnosed and identifying those at risk for severe disease remains challenging. The 2018 American Heart Association guidelines recommend ECG and cardiac rhythm strip, but there is no consensus on the approach to further cardiac evaluation in the US or management of Chagas cardiomyopathy. This study retrospectively analyzed symptoms and cardiac examination of migrants predominantly from endemic Central American countries who were diagnosed with Chagas disease and received care at Boston Medical Center. Most patients were asymptomatic and nearly half had evidence of cardiomyopathy. Echocardiogram abnormalities were present in approximately one-third of asymptomatic individuals with normal ECGs. Some individuals had significant abnormalities found on heart rhythm monitors or cardiac magnetic resonance imaging which led to management changes. These findings underscore the importance of routine echocardiograms, consideration of cardiac rhythm monitoring, and thorough cardiac assessment in people with Chagas disease regardless of symptoms.

## Introduction

Chagas disease is the most common neglected tropical disease in the United States (US) with approximately 300,000 people living with infection [1]. Infection with *Trypanosoma cruzi*, the parasite that causes Chagas disease, primarily occurs in Mexico, Central, and South America. Approximately 70% of individuals remain asymptomatic and in the chronic indeterminate stage, however, up to 30% progress to the determinate form, characterized by potentially life-threatening cardiac or gastrointestinal complications [2]. The number of affected individuals in the US and other non-endemic countries may increase with current migration trends and cases of congenital and autochthonous vector transmission have been demonstrated in the US [1,3]. Despite the prevalence and substantial long-term sequelae of Chagas disease, most US

healthcare providers are not familiar with the approach to its diagnosis or management; an estimated less than 1% of cases are diagnosed [1,4–6]. As a result of this systematic underdiagnosing, there are limited data on Chagas disease prevalence and cardiac disease burden amongst people living in the US.

Cardiac manifestations of Chagas disease usually begin in early adulthood and progress over decades [2]. Initial presentation often includes conduction abnormalities on electrocardiogram (ECG). Long-term sequelae include complex ventricular arrhythmias, sinus node dysfunction and severe bradycardia, high degree atrioventricular block, thromboembolic disease, progressive dilated cardiomyopathy with congestive heart failure, and sudden cardiac death [2,7,8]. Benznidazole and nifurtimox are the current antiparasitic treatment options for Chagas disease [2]. Early antitrypanosomal treatment may prevent Chagas cardiomyopathy development for patients in the indeterminate stage [9]. In contrast, in patients with established cardiomyopathy, current data suggest that antiparasitic treatment does not seem to prevent cardiomyopathy progression [8] and cardiac treatment options are focused on standard heart failure therapies [2,10,11]. The risk of potentially life-threatening cardiac complications highlights the importance of early diagnosis of *T. cruzi* infection.

Chagas disease is increasingly recognized as an important cause of non-ischemic cardiomyopathy (NICM) and stroke in the US. Studies of migrants from Latin America living in Los Angeles, Washington D.C., and New York City found similar rates (13–19%) of NICM attributed to Chagas disease [12–14]. In the US, Chagas cardiomyopathy is associated with at least three times higher mortality compared to non-Chagas cardiomyopathy, as well as increased cardiomyopathy-related hospitalizations compared to non-Chagas cardiomyopathy [12]. Patients with Chagas disease have increased risk of thromboembolic disease and stroke, and mortality from stroke in patients who are *T. cruzi* seropositive may be higher compared to those who are seronegative [15].

Based on other successful US Chagas disease screening programs, the Strong Hearts project integrated Chagas screening into primary care at East Boston Neighborhood Health Center (EBNHC) in 2017 [16,17]. They demonstrated a 0.97% screening positivity rate at EBNHC, similar to other published Chagas screening program results [17,18]. Referrals from EBNHC to the Boston Medical Center (BMC) Center for Infectious Disease (CID), as well as efforts to promote screening across the BMC hospital system, have identified a large cohort of patients living with Chagas disease in Boston. The aim of this paper is to describe the cardiac manifestations and disease burden of Chagas disease and determine the prevalence of cardiac disease among asymptomatic individuals who underwent evaluation and management in the BMC CID.

## Methods

### Study setting, protocol, and data collection

We conducted a retrospective descriptive analysis of patients aged 18 years or older with Chagas disease seen at the CID from January 2016 to January 2023. Starting in 2017, one-time Chagas screening was recommended by the Strong Hearts project in all patients at EBNHC less than 50 years old who had lived in Mexico, South or Central America for at least 6 months, in accordance with expert recommendations [19]. Patients whose initial positive Chagas screening ELISA test performed by a commercial laboratory was positive had a confirmatory test performed at the Center for Disease Control and Prevention (CDC) [the Wiener Chagatest recombinante v.3.0 enzyme immunoassay (EIA; Wiener Laboratories, Rosario, Argentina) or the trypomastigote excreted-secreted antigens (TESA)-blot, with discordant results mediated by an immunofluorescence assay [20]. Patients with two different positive serologic tests were

referred to the CID for further evaluation and treatment. Patients with one initial positive serology who had either a negative second serology or did not have a second confirmatory serology, were excluded from this analysis.

Chart review was performed for demographic data, medical comorbidities, and substance use (including alcohol or tobacco use for greater than one year). Symptoms and clinical manifestations of Chagas disease were assessed by review of infectious disease physicians' and cardiologists' notes in the electronic medical record.

Standard clinical practice for Chagas disease at the CID includes baseline ECG, blood chemistries, complete blood count, and cardiac biomarkers (troponin, brain natriuretic peptide [BNP]). Baseline transthoracic echocardiogram (TTE) was obtained in most patients; at the start of the study, patients who were asymptomatic with normal ECGs did not necessarily have TTE performed, but this practice changed over time so that all patients received a baseline echocardiogram. Contrast was requested for all TTEs to improve visualization of the cardiac apex [21]. If patients underwent further cardiac evaluation or intervention (e.g., ambulatory cardiac monitoring, exercise tolerance [ETT] or pharmacologic stress testing, cardiac magnetic resonance imaging [CMR], cardiac catheterization, pacemaker or implantable cardioverter defibrillator [ICD] placement) this information was collected. Quantification of the burden of late gadolinium enhancement (LGE) in the CMRs was performed by a cardiologist with a semi-automated technique using Circle Cardiovascular Imaging software (Calgary, Alberta) [22].

The data presented in this study are presented in aggregate without any identifying information and individual consent was not obtained. Research approval was obtained by the Institutional Review Board of Boston Medical Center and community board approval from East Boston Neighborhood Health Center (H-39646).

## Study definitions

Patients were considered to have been screened for Chagas disease if their physician ordered the test based on migration history or country of origin. Assessment of symptoms was based on review of patients' medical records after infectious disease and cardiology evaluation. Any patient endorsing symptoms or clinical manifestations potentially attributed to Chagas disease, including cardiac, gastrointestinal, and/or neurologic features, was classified as symptomatic. Stroke was included as a clinical manifestation of infection based on prior studies [23]. Asymptomatic patients denied any of the symptoms or clinical features above.

Chagas cardiomyopathy stage was determined for each patient according to the American Heart Association definitions [2] (S1 Table). The Rassi score, a metric comprised of 6 independent parameters associated with mortality risk for patients with Chagas disease, was calculated for patients who had all 6 parameters available [24] (S2 Table).

## Analysis

Descriptive analysis was stratified based on the presence or absence of symptoms (without incorporating ECG findings) to evaluate the cardiac manifestations in these two clinically relevant groups. Due to the exploratory nature of the study and the absence of specific hypotheses, as well as the small sample size, data are reported as trends without p-values. Standard deviation was calculated for the mean age of patients in the study.

## Results

A total of 109 patients with Chagas disease were identified; 86 (79%) were detected through screening. Among the 86 patients who were screened, 39 (45%) of tests were ordered by

internists, 36 (42%) by family medicine physicians, 10 (12%) by obstetricians, and 1 (1%) by the American Red Cross during blood donation. Chagas serology was obtained due to clinical suspicion for the remaining 23 patients by cardiologists (16/23, 70%), internists (5/23, 22%), and family medicine physicians (2/23, 9%).

### Demographics and underlying comorbidities

Most individuals diagnosed with Chagas disease were female (67, 61%); 22% of women were of child-bearing age (range 22–40 years old) and 15% of women were pregnant. Mean age was 43 years with 27% of patients > 50 years of age. Most were migrants from Central America (95%); 93 individuals (85%) were from El Salvador. Many patients had cardiovascular disease risk factors including obesity, hypertension, hyperlipidemia, diabetes, and coronary artery disease (**Table 1**).

### Clinical features and symptoms

Among all patients, the most common symptoms of cardiac disease were palpitations (25%), chest pain (17%), and dyspnea on exertion (14%); gastrointestinal complaints, including heartburn, constipation, abdominal pain, and/or dysphagia were also noted in up to 13 patients (12%). Neurologic symptoms included dizziness or lightheadedness (7%) and headache (5%). Stroke was diagnosed in 4 patients (4%), including embolic stroke in 3 patients and hemorrhagic stroke in 1 patient. Notably, two of the patients who experienced strokes were women less than 40 years old without alternative risk factors for stroke other than left ventricular aneurysms identified on both patients' echocardiograms; one patient was noted to have atrial fibrillation before her stroke. Over half (57, 52%) of the 109 patients were considered asymptomatic (**Table 1**). Among patients ages 18–30 years, 16/22 (73%) were asymptomatic; however, among patients over age 50, less than half (14/29, 48%) were asymptomatic.

### Cardiac findings in patients with Chagas Disease (Tables 2 and 3)

**Biomarkers.** Troponin was above the upper limit of normal in 5 of 54 (9%) patients for whom troponin testing was done; all had symptoms or clinical manifestations of disease (**Table 2**). Elevated BNP (>100 pg/mL) was seen in 10 of 43 (23%) patients tested; one was asymptomatic.

### Electrocardiogram

ECG tracings were normal in 52/102 (51%) patients, of whom 32% had symptoms or clinical manifestations of disease. ECG abnormalities were more common in those >50 years old compared to those <50 years old (20/29 or 70% vs 30/73 or 41%), but notably were observed in 12/53 (23%) of those <40 years old (**S3 Table**). The most common ECG abnormalities were complete/incomplete RBBB, LAFB, and T-wave abnormalities. Combined RBBB and LAFB (10%), low voltage (12%), and multiple ECG abnormalities (27%) were seen more frequently in symptomatic compared to asymptomatic individuals (**Table 2**).

### Transthoracic echocardiogram

Baseline TTEs were obtained in 94 of 109 (86%) patients. TTE abnormality was noted in 49% of individuals with TTE, including 23/44 (52%) asymptomatic patients, 13/42 (31%) patients with normal ECG (**Table 2**), and 25/64 (39%) patients <50 years old. Common TTE abnormalities in the 94 patients included left atrial dilation (21%), diastolic dysfunction (17%), and apical hypokinesis (11%). While 89% of patients had a normal left ventricular ejection fraction

**Table 1. Demographic and Clinical Characteristics.**

| | Asymptomatic (n = 57) | Symptomatic (n = 52) | Total (n = 109) |
|---|---|---|---|
| **Demographics** | | | |
| Mean age, years (SD) | 38 (13) | 44 (13) | 43 (13) |
| Age range, years–no. (%) | | | |
| 18–30 | 16 (28) | 6 (12) | 22 (20) |
| 31–40 | 13 (23) | 13 (25) | 26 (24) |
| 41–50 | 14 (25) | 18 (35) | 32 (29) |
| >50 | 14 (25) | 15 (29) | 29 (27) |
| Sex, female—no. (%) | 33 (58) | 34 (65) | 67 (61) |
| Country of origin—no. (%) | | | |
| El Salvador | 49 (86) | 44 (85) | 93 (85) |
| Guatemala | 3 (5) | 3 (6) | 6 (6) |
| Honduras | 2 (4) | 3 (6) | 5 (5) |
| Colombia | 1 (2) | 1 (2) | 2 (2) |
| Bolivia | 1 (2) | 0 (0) | 1 (1) |
| Brazil | 0 (0) | 1 (2) | 1 (1) |
| Mexico | 1 (2) | 0 | 1 (1) |
| Primary language—no. (%) | | | |
| Spanish | 57 (100) | 51 (98) | 108 (99) |
| Portuguese | 0 | 1 (2) | 1 (1) |
| **Medical Comorbidities—no. (%)** | | | |
| Hypertension | 11 (19) | 23 (44) | 34 (31) |
| Hyperlipidemia | 13 (23) | 15 (29) | 28 (26) |
| Coronary artery disease | 0/47 | 6/50 (12) | 6/97 (6) |
| Diabetes mellitus | 6/41 (15) | 7/41 (17) | 13/82 (16) |
| Overweight | 34 (60) | 20 (38) | 54 (50) |
| Obese | 20 (35) | 27 (52) | 47 (43) |
| Chronic kidney disease | 0/55 | 4/52 (8) | 4/107 (4) |
| Tobacco use* | 10 (18) | 10 (19) | 20 (18) |
| Alcohol use** | 18 (32) | 16 (31) | 34 (31) |

Continuous variable expressed as mean (SD); categorical variable expressed as n (%)

Incomplete data variables are indicated with denominators.

Coronary artery disease is defined as atherosclerotic disease identified by left heart catheterization or listed in the electronic medical record as a medical problem.

Diabetes Mellitus is defined as hemoglobin A1c > 6.5%.

Overweight, defined as body mass index between 25–30 kg/m2; obese, defined as body mass index > 30 kg/m2.

Chronic kidney disease, defined as GFR < 60 mL/min/1.73 m2 on at least two separate measurements at least 6 months apart.

*Any amount of tobacco use for more than one year.

**Any amount of alcohol consumption.

(LVEF) > 50%, 11% of individuals had LV systolic dysfunction with LVEF ≤ 50%. LV apical abnormalities were detected in 11% of patients.

## Ambulatory cardiac monitors

Ambulatory cardiac monitoring was done in 24 patients (**Table 3**). Monitoring was performed for 30 days in 16 (67%) patients and for 1–14 days in the remaining 8 patients (33%). Ventricular arrhythmias were present in 50% of event monitors placed, including premature ventricular contractions ([PVC] 38%, with PVC burden ranging from rare to 8% of the monitoring

**Table 2. Cardiac Findings in Chagas Disease.**

| | Asymptomatic (n = 57) | Symptomatic (n = 52) | Total (n = 109) |
|---|---|---|---|
| **Cardiac Biomarkers–no. (%)** | | | |
| Abnormal troponin I | 0/21 (0) | 5/33 (15) | 5/54 (9) |
| Abnormal BNP | 1/19 (5) | 9/24 (38) | 10/43 (23) |
| **Baseline ECG* - no. (%)** | **n = 51** | **n = 51** | **n = 102** |
| Normal sinus rhythm | 37 (73) | 15 (30) | 52 (51) |
| RBBB | 4 (8) | 15 (30) | 19 (19) |
| T-wave abnormalities | 7 (14) | 11 (22) | 18 (18) |
| LAFB | 2 (4) | 9 (18) | 11(11) |
| Sinus bradycardia | 5 (10) | 4 (8) | 9 (9) |
| PACs/PVCs | 3 (6) | 6 (12) | 9 (9) |
| Low voltage | 1 (2) | 6 (12) | 7 (7) |
| RBBB/LAFB | 1 (2) | 5 (10) | 6 (6) |
| Left axis deviation | 1 (2) | 3 (6) | 4 (4) |
| Atrial fibrillation/flutter | 0 (0) | 4 (8) | 4 (4) |
| Second degree AV block | 0 (0) | 3 (6) | 3 (3) |
| Prolonged QT interval | 1 (2) | 2 (4) | 3 (3) |
| Multiple abnormalities | 6 (12) | 14 (27) | 20 (20) |
| No ECG performed | 6 (11) | 1 (2) | 7 (6) |
| **Baseline TTE* - no. (%)** | **n = 44** | **n = 50** | **n = 94** |
| Normal | 21 (48) | 27 (54) | 48 (51) |
| Abnormal | 23 (52) | 23 (49) | 46 (49) |
| *Ventricular abnormalities* | | | |
| LVEF > 50% | 40 (91) | 44 (88) | 84 (89) |
| LVEF 40–50% | 3 (7) | 3 (6) | 6 (6) |
| LVEF < 40% | 1 (2) | 3 (6) | 4 (4) |
| Diastolic dysfunction | 9 (20) | 7 (14) | 16 (17) |
| Apical hypokinesis | 4 (9) | 6 (12) | 10 (11) |
| Right ventricular dilation | 5 (11) | 4 (8) | 9 (10) |
| Left ventricular dilation | 2 (5) | 5 (10) | 7 (7) |
| Apical aneurysm | 2 (5) | 5 (10) | 7 (7) |
| Apical thrombus | 1 (2) | 2 (4) | 3 (3) |
| *Atrial Abnormalities* | | | |
| Left atrial dilation | 9 (20) | 11 (22) | 20 (21) |
| Right atrial dilation | 1 (2) | 4 (8) | 5 (5) |
| Moderate-severe valvular disease | 3 (7) | 5 (10) | 8 (9) |
| Pericardial effusion | 3 (7) | 0 | 3 (3) |
| **ECG normal, TTE abnormal** | **11/28 (39)** | **2/14 (14)** | **13/42 (31)** |

*ECG or TTE collected in closest proximity to Chagas disease diagnosis; some patients had more than one ECG or TTE abnormality, so these percentages do not add up to 100%

Abnormal troponin (defined as greater than the upper limit of normal; the upper limit of normal differed by testing type i.e. high sensitivity versus traditional troponin analyses)

Abnormal b-type natriuretic peptide (BNP), value >100 pg/mL)

BNP = b-type natriuretic peptide; PAC= premature atrial contraction; PVC=premature ventricular contraction; RBBB = right bundle branch block; LAFB = left anterior fascicular block; AV = atrio-ventricular; ECG = electrocardiogram; TTE = transthoracic echocardiogram; LVEF = left ventricular ejection fraction

**Table 3. Advanced Cardiac Diagnostics and Interventions in Patients with Chagas.**

| | Asymptomatic (n = 57) | Symptomatic (n = 52) | Total (n = 109) |
|---|---|---|---|
| **Cardiac Event Monitor—no. (%)** | **n = 2** | **n = 22** | **n = 24** |
| Ventricular arrhythmias: | 2 (100) | 10 (45) | 12 (50) |
| PVCs | 2 (100) | 7 (32) | 9 (38) |
| Non-sustained ventricular tachycardia | 0 (0) | 5 (23) | 5 (21) |
| Polymorphic ventricular tachycardia | 0 | 1 (5) | 1 (4) |
| Atrial arrhythmias: | 1 (50)* | 4 (18) | 5 (21) |
| Atrial fibrillation | 1 (50) | 2 (9) | 3 (13) |
| Premature atrial contraction | 1 (50) | 2 (9) | 3 (13) |
| Sinus bradycardia | 1 (50) | 3 (14) | 4 (17) |
| Sinus pauses | 0 | 2 (9) | 2 (8) |
| **Stress Test[†] - no. (%)** | **n = 8** | **n = 19** | **n = 27** |
| Chronotropic incompetence | 1 (13) | 5 (26) | 6 (22) |
| Exercise-Induced Ischemia | 2 (25) | 3 (16) | 5 (19) |
| Exercise-Induced PVCs | 1 (13) | 3 (16) | 4 (15) |
| **Cardiac MRI—no. (%)** | **n = 3** | **n = 9** | **n = 12** |
| LGE presence | 2 (67) | 4 (44) | 6 (50) |
| LGE 1–10%** | 2 (67) | 2 (22) | 4 (33) |
| LGE >10% | 0 | 2 (22) | 2 (17) |
| Unable to assess | 0 | 2 (22) | 2 (17) |
| Apical aneurysm not noted on TTE | 1 (33) | 0 (0) | 1 (8) |
| Apical clot not noted on TTE | 0 (0) | 1 (11) | 1 (8) |
| Apical hypokinesis not noted on TTE | 0 (0) | 1 (11) | 1 (8) |
| **Cardiac Catheterization–no. (%)** | **n = 0** | **n = 10** | |
| Nonobstructive CAD | 0 (0) | 3 (30) | |
| Obstructive CAD | 0 (0) | 3 (30) | |
| **Pacemaker/ICD Placement—no. (%)** | **0/61** | **7/48 (15)** | **7/109 (6)** |

PVCs, premature ventricular contractions; MRI, magnetic resonance imaging; LGE, late gadolinium enhancement; TTE, transthoracic echocardiogram; CAD, coronary artery disease; ICD, implantable cardioverter-defibrillator

*One unique asymptomatic patient had both atrial fibrillation and premature atrial contractions at different times during cardiac ambulatory monitoring.

[†]Stress tests constitute exercise-tolerance tests and pharmacologic stress tests

**% LGE calculated by the LGE volume (in grams) per total myocardial mass (in grams)

time) and non-sustained ventricular tachycardia (NSVT) (21%). Other arrhythmias included atrial arrhythmias (21%), sinus bradycardia (17%), and sinus pauses (8%). One patient had nocturnal and diurnal sinus pauses that were asymptomatic and between 3–7 seconds long; this information was unavailable for the other patient with sinus pauses. Arrhythmias were identified in 3 patients <40 years old.

## Exercise tolerance/cardiac stress testing

Exercise tolerance tests (ETT) or pharmacologic cardiac stress test were performed in 27 patients (**Table 3**). Among ETTs, 22% demonstrated chronotropic incompetence, or the inability to increase heart rate adequately during exercise to match cardiac output to metabolic demands. Among patients with chronotropic incompetence, 5 out of 6 (83%) had symptoms including palpitations, chest pain, and/or dyspnea. Exercise-induced PVCs were noted in 15% of patients. Exercise-induced ischemia was seen in 5 patients (19%); of these 5 patients, 2 had left-heart catheterization demonstrating obstructive CAD and 3 had stress tests demonstrating

partially reversible perfusion abnormalities. Symptoms of heart disease were present in 70% of patients with ETT or pharmacologic stress testing. Among patients without symptoms of cardiac disease who underwent stress testing, 50% had chronotropic incompetence, exercise-induced PVCs, or exercise-induced ischemia.

## Cardiac MRI (CMR)

Twelve patients had CMR performed; of these, two patients had apical structural abnormalities identified on CMR but not detected on TTE (1 apical aneurysm, 1 LV apical hypokinesis/thrombus). Both patients were subsequently started on therapeutic anticoagulation. LGE was evaluated in ten CMRs and observed in 6/10 (60%) of those with LGE analysis available (**Table 3**). Most patients had between 1–10% LGE; two symptomatic patients had >10% LGE (**Table 3**). Notably, 2 out of 3 (67%) asymptomatic patients with CMR available had LGE. CMRs were ordered by cardiologists in 3 asymptomatic individuals to investigate newly noted reduced LVEF on TTE (2/3, 67%) or, in one patient with normal LVEF, to assess degree of cardiac fibrosis while considering antiparasitic therapy (1/3, 33%). Most patients with LGE on CMR were in early stages of cardiomyopathy (4/6 patients in stage B1), while 2 were in more advanced stages (1 in B2, 1 in C). Their ECGs showed RBBB (2/6), T-wave inversions only (2/6), atrial fibrillation and T-wave inversions (1/6), PVCs and left atrial dilation (1/6).

## Cardiac catheterization

Ten symptomatic patients underwent left-sided cardiac catheterization (**Table 3**). Three patients (30%) had obstructive CAD requiring interventional techniques. Seven (70%) had normal cardiac catheterization or non-obstructive CAD, suggesting that Chagas disease was the etiology of their cardiomyopathy. Other typical nonischemic cardiomyopathy etiologies were ruled out for these patients (including thyroid dysfunction, iron studies, and human immunodeficiency virus) and the diagnosis of nonischemic cardiomyopathy due to Chagas disease was a diagnosis of exclusion in the setting of positive serologies.

## Pacemaker/Implantable Cardioverter Defibrillator (ICD)

Seven symptomatic patients (6%) required pacemaker or ICD placement, including 2 who were <50 years old (**Table 3**). Reasons for pacemaker placement included sinus pauses (1/7, 14%), symptomatic sinus node dysfunction (1/7, 14%), and unknown due to device placement in another country (1/7, 14%). ICDs were placed for primary prevention given significantly reduced EF (3/7, 43%) and ventricular arrhythmia (1/7, 14%). Two patients who underwent ICD placement during the study for primary prevention developed asymptomatic malignant arrhythmias remotely recorded by their ICDs that led to hospital admission for medical cardioversion or ablation. The first patient was intolerant to three anti-arrhythmic medications (amiodarone, sotalol, and dofetilide), developed refractory atrial fibrillation despite pulmonary vein isolation that was revised twice, and was admitted several times for cardioversion. The second patient was admitted for ablation after receiving two asymptomatic shocks at home for rapidly conducting supraventricular tachycardia.

## Cardiac abnormalities requiring clinical intervention

After Chagas disease diagnosis, cardiac evaluation led to clinical findings for which a therapeutic intervention was necessary (**Fig 1**). Fifty-two intervenable findings were found in 25 (23%) unique patients, with some individuals having multiple abnormalities. Echocardiograms resulted in changes in medical management for 15 (16%) patients; 7 were <50 years old and 3

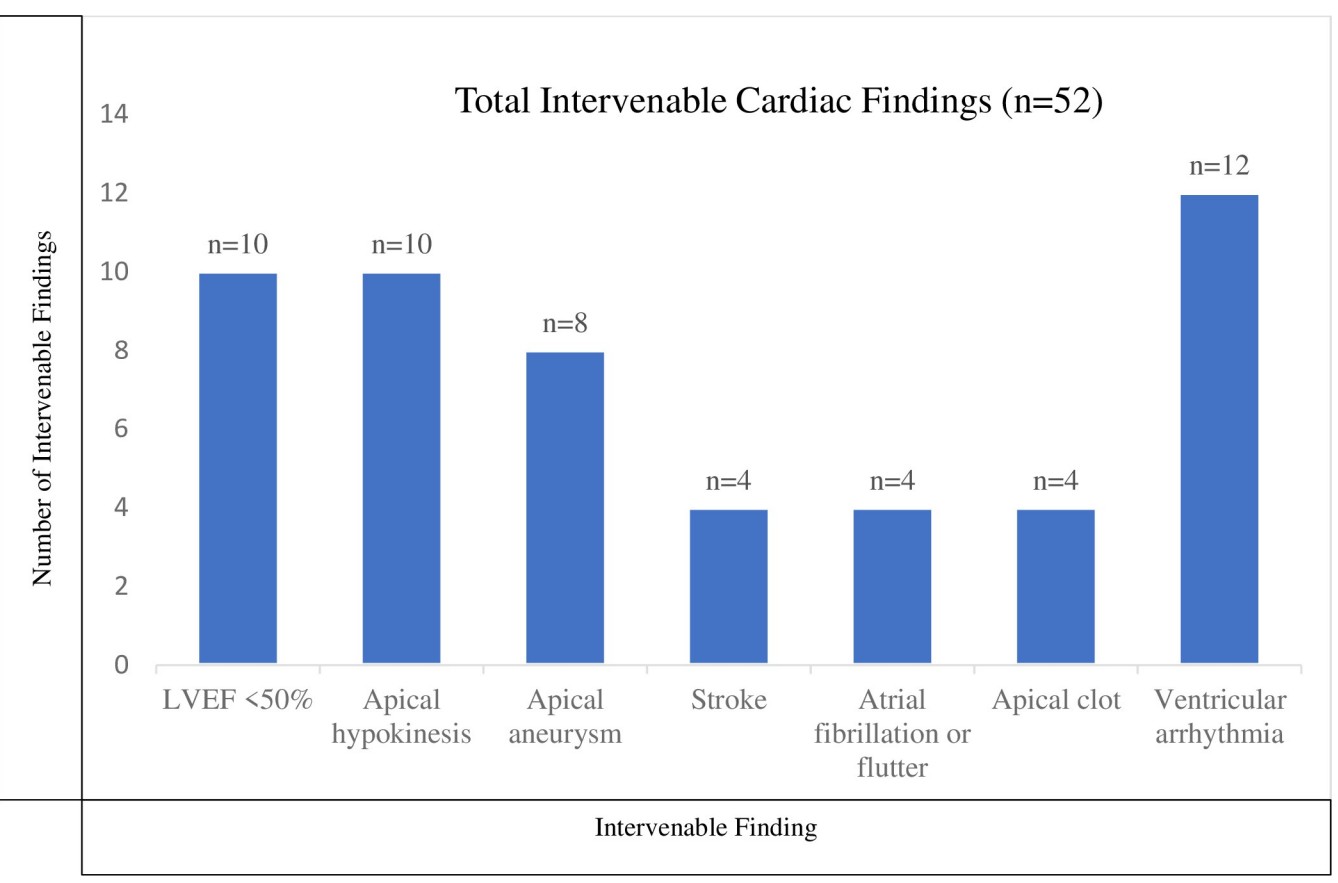

**Fig 1. Cardiac Findings Requiring Clinical Intervention.** *52 cardiac findings noted in 25 unique patients after workup for Chagas disease. This figure demonstrates the types of cardiac abnormalities associated with Chagas disease found in patients after they were diagnosed with Chagas disease and underwent cardiac evaluation.

were <40 years old. In several cases, multiple therapeutic changes were implemented for a single individual. Eleven (12%) patients were started on angiotensin-converting enzyme inhibitor, or angiotensin receptor blocker, 8 (9%) on beta-blocker, 6 (6%) on therapeutic anticoagulation, and 2 (2%) on low-dose aspirin (**Fig 2**). Ten patients had potentially intervenable findings that did not result in management change: one patient declined the management change, 3 patients did not require management change (i.e. found to have apical hypokinesis but was already on anticoagulation for another indication) and 6 did not have a reason documented. Cardiac monitoring results led to change in management in 5 (21%) patients; 1 patient was initiated on a beta-blocker for frequent PVCs, 2 had their beta-blocker dose increased for NSVT, 1 was started on therapeutic anticoagulation for paroxysmal atrial fibrillation, and 1 had a pacemaker placed for multiple sinus pauses (**Fig 2**). Three ICDs were placed during the study period subsequent to Chagas disease cardiac evaluation, including one in a patient who was 36 years old (**Figs 1 and 2**).

## Chagas cardiac classification systems

AHA stage of Chagas cardiomyopathy was identified for the 94 patients who had TTE and ECG results; 48 (51%) were stage A, 30 (32%) stage B1, 7 (7%) stage B2, 8 (9%) stage C, and 1 (1%) stage D (**Tables 4 and S1 for AHA classification**). Patients without cardiomyopathy

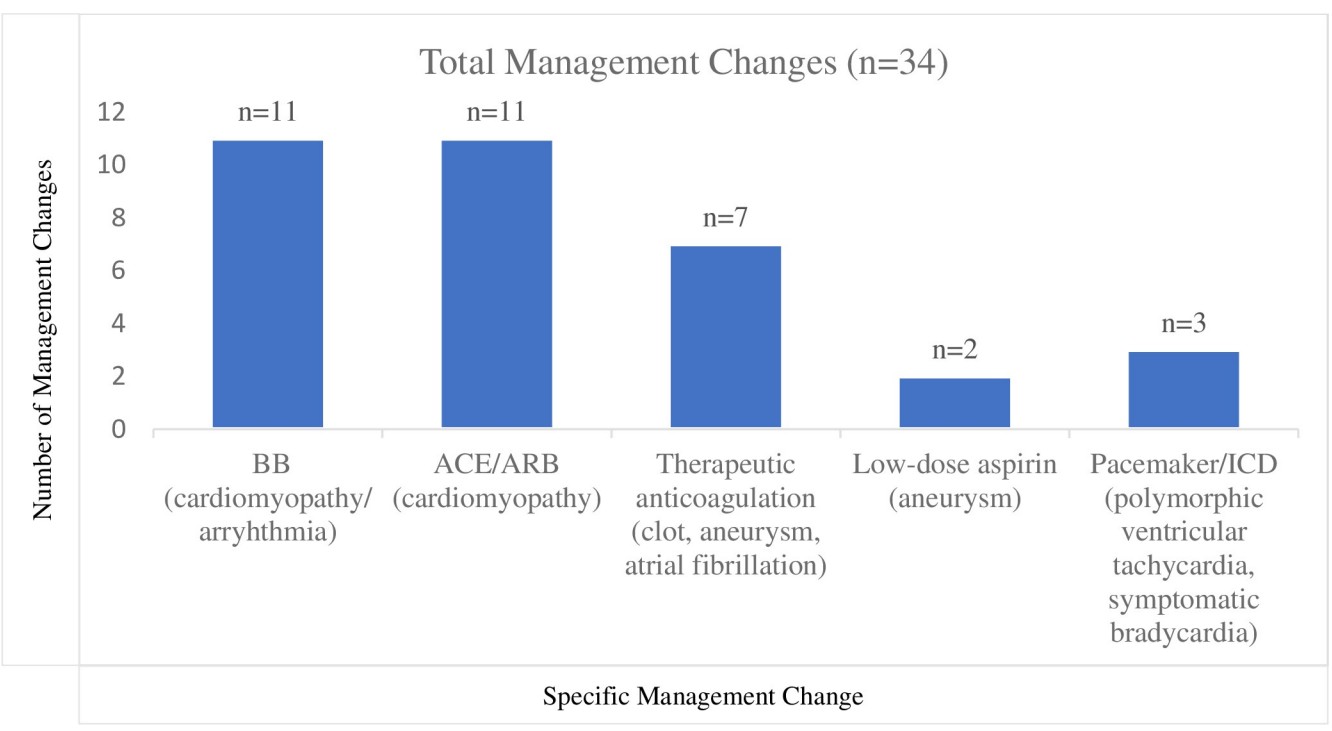

**Fig 2. Clinical Management Changes Due to Chagas Disease Cardiac Evaluation.** *In 15 individual patients. BB, beta-blocker; ACE, angiotensin-converting enzyme inhibitor; ARB, angiotensin receptor blockers; ICD, implantable cardioverter-defibrillator. This figure demonstrates the cardiac clinical management changes that occurred after Chagas disease diagnosis and cardiac evaluation uncovered disease that required intervention.

(stage A) tended to be younger with 42/48 (88%) <50 years old, while 22/46 (48%) of those with cardiomyopathy (stages B1-D) were <50 years old. However, 8 patients who were less than 40 years old had cardiomyopathy of stage B1 or worse (**S4 Table**). Of the 24 patients who underwent ambulatory cardiac monitoring, 79% were in the Rassi low-risk category, 8% medium risk, and 13% high-risk, which are associated with a 10-year mortality risk of 10%, 44%, and 84%, respectively (**Table 4**).

**Table 4. American Heart Association Class and Rassi Score.**

| American Heart Association Chagas Cardiomyopathy (CCM) Stage—no. (%) | | | |
|---|---|---|---|
| | **Asymptomatic** (n = 44) | **Symptomatic** (n = 50) | **Total** (n = 94)* |
| A (Indeterminate form) | 29 (66) | 19 (38) | 48 (51) |
| B1 (Structural CCM) | 10 (23) | 20 (40) | 30 (32) |
| B2 (Chagas dilated CM/heart failure, asymptomatic) | 4 (9) | 3 (6) | 7 (7) |
| C (Chagas dilated CM/heart failure, symptomatic) | 1 (2) | 7 (14) | 8 (9) |
| D (Chagas dilated CM/heart failure, poorly controlled) | 0 | 1 (2) | 1 (1) |
| Rassi Criteria Category—no (%) | | | |
| | **Asymptomatic** (n = 2) | **Symptomatic** (n = 22) | **Total** (n = 24)† |
| Low (0–6 points) | 2 (100) | 17 (77) | 19 (79) |
| Medium (7–11 points) | 0 | 2 (9) | 2 (8) |
| High (12–20 points) | 0 | 3 (14) | 3 (13) |

Refer to **S1** and **S2** Tables regarding AHA classification and Rassi score details

*Unable to determine AHA class for 15 patients given missing TTE

†Rassi score calculated in 24 patients who had both ambulatory cardiac monitor and TTE

## Discussion

This is one of the largest case series describing the cardiac manifestations of Chagas disease in the US with 109 patients with confirmed Chagas disease. Over half of patients were asymptomatic and most patients (79%) were screened by a health care provider, rather than being tested based on symptomatology or disease features. Over half of the asymptomatic patients had abnormal ECG and/or echocardiogram findings, highlighting the importance of cardiac evaluation for individuals with Chagas disease regardless of symptoms. Echocardiographic abnormalities were seen in 31% of patients with normal ECGs and led to change in management in 16% of patients, suggesting that echocardiography should be routine and, ideally, include contrast [25].

The current standards for monitoring Chagas cardiomyopathy are ECG and echocardiogram. Our findings corroborate other studies that describe patients with Chagas disease who have normal ECGs and yet they have echocardiographic changes, including a recent Spanish study where 10.8% of patients with Chagas cardiomyopathy had normal ECGs [25, 26]. Echocardiography techniques such as calculation of left ventricle strain or the BENEFIT trial's wall motion score may better predict development of Chagas cardiomyopathy or cardiovascular events, respectively, but these are not routinely performed [27, 28]. These are useful for monitoring heart failure progression but do not capture arrhythmias. Sudden cardiac death (SCD) can occur without any prior ECG abnormalities or symptoms [2,10,29]. Studies have shown that self-limited ventricular arrhythmias in Chagas cardiomyopathy are not random phenomena and monitoring for only 24 hours may be insufficient to detect arrhythmia burden [29]. Ambulatory cardiac monitoring uncovered ventricular arrhythmia in 50% of patients who underwent monitoring in our study, including two asymptomatic patients, indicating a high burden of arrhythmia overall. For 5 patients in our study, ambulatory cardiac monitoring prompted by the Chagas diagnosis led to change in clinical management. As an example, one patient without cardiac symptoms who presented following a stroke was found on subsequent workup to have malignant arrhythmias prompting ICD placement. Two patients developed malignant arrhythmias detected remotely by their ICDs, despite not having any symptoms at the time, which led to hospital admission for treatment. This suggests that ambulatory monitoring should be strongly considered for many patients with newly diagnosed Chagas disease, irrespective of symptoms. While the underlying immunopathological mechanism of Chagas cardiomyopathy makes it challenging to predict disease course, extended cardiac monitoring may provide additional clues about individuals' risk and guide therapies, such as beta-blocker initiation, in patients with ventricular arrhythmias.

The use of advanced imaging modalities, such as CMR or positron emission tomography (PET), is limited by their cost and availability. However, compared to TTE, CMR has improved sensitivity to detect early myocardial fibrosis via LGE, which is associated with high risk of sudden cardiac death from malignant arrhythmia [30, 31]. In our study, 2 out of 3 asymptomatic patients who underwent CMR had LGE presence in a pattern consistent with Chagas cardiomyopathy [25, 32]. Both mid-wall and epicardial hyperenhancement patterns often in a focal pattern can be supportive of Chagas cardiomyopathy by CMR [32]. Early detection of LGE may have implications for clinical care, including consideration of ICD, particularly when associated with ventricular arrhythmias or reduced ejection fraction. Despite the use of echocardiographic contrast, TTE can be limited by inadequate windows. CMR may provide better detection of advanced stages of disease, including regional wall motion abnormalities, apical aneurysm, and intracardiac thrombi [25]. One asymptomatic patient in our study ultimately was found to have an apical aneurysm and clot on CMR that was not seen on TTE, changing the course of her clinical care. No patient in our study underwent PET-CT and the

use of PET-CT in Chagas disease to detect pathologic inflammation has not been systematically studied. However, three case reports have described pathologic uptake on PET-CT in patients with Chagas disease and ventricular tachycardia, suggesting that inflammation in these areas may contribute to arrhythmia and identify targets for ablative procedures [33,34]. While not recommended for all people living with Chagas disease, these advanced imaging modalities may improve risk stratification for some patients and have the potential to inform treatment.

Over half (51%) of our patients were in the indeterminate stage A of disease, without evidence of cardiac involvement, and, thus, they may benefit from antitrypanosomal treatment to prevent progression to cardiomyopathy [9]. While a notable proportion (32%) were identified at an early stage of cardiomyopathy, further studies are needed to understand if antitrypanosomal therapy is beneficial for these patients and for patients 50 years or older. Unfortunately, for the remaining 17% of our patients in more advanced stages of cardiomyopathy, antitrypanosomal therapy does not seem to prevent death or major cardiovascular events [8]. This underscores the importance of identifying individuals early to reduce progression to cardiomyopathy, which requires screening individuals at risk for Chagas disease and close collaboration between infectious disease and cardiology physicians. In patients with more advanced disease, identification of Chagas disease has other important implications. We identified life-threatening complications of Chagas disease, including stroke, arrhythmia, and ventricular apical aneurysm and thrombus, which allowed for more intensive monitoring and advanced management. Overall, 9% of patients had cardiac apical aneurysm, including 2 patients less than 40 years old, and 4% had apical clot. Three patients, including a patient under 40 years old, had an ICD placed after Chagas diagnosis that may have prevented sudden death [11]. Additional benefits of more intensive cardiology management for patients with Chagas cardiomyopathy include identifying patients who may benefit from cardioembolic stroke prophylaxis, and optimization of heart failure medications [15].

This study has several limitations. It is a single-site, retrospective analysis and we can report trends, but, for the most part, is not large enough to draw statistical comparisons. Symptoms were obtained from electronic medical record documentation and may be inconsistently reported by patients and/or documented. Echocardiography techniques to calculate left ventricular strain and the wall motion score were not performed prospectively. The decision to pursue CMR or event monitor was driven by clinical evaluation, which may overestimate the proportion of abnormal findings. Our data may not be representative of Chagas disease manifestations in all migrant groups. There are at least six *T. cruzi* discrete type units (DTUs) with different geographic variabilities [35]. The majority (85%) of patients in our study were from El Salvador, where Tc1 may be the most common circulating strain and may be associated with different clinical features than other DTUs [35]. Gastrointestinal involvement of Chagas disease and severe cardiac manifestations are more common in the Southern Cone of South America and may be underrepresented in our cohort of predominantly Central American origin [35]. Our study was also impacted by the COVID-19 pandemic, which further limited access to primary care services in the at-risk population and likely led to decreased screening.

This study demonstrates a significant burden of Chagas cardiac disease amongst migrants living in a non-endemic area, the majority of whom were asymptomatic. We demonstrate that normal ECG does not equate to absence of cardiac disease. Additional testing with echocardiography, ambulatory cardiac monitoring, and CMR often detects occult, subclinical cardiac disease which drives changes in clinical management. We identified patients in the indeterminate stage who may benefit from antiparasitic therapy to minimize risk of disease progression. Our experience has shown that early identification and intervention is particularly crucial given progression to end-stage heart failure can be particularly challenging for individuals

with Chagas disease, many of whom are already wrestling with economic, social, and immigration vulnerabilities. While some patients sought medical care for their symptoms related to Chagas disease, many were asymptomatic and did not. Better understanding the barriers to care that this vulnerable population faces will be key to improving the health of people living with Chagas disease. As we continue to improve Chagas disease diagnostics and increase awareness amongst providers and patients about Chagas disease, screening individuals at risk remains critical. Our findings also highlight the importance of Chagas disease testing for symptomatic individuals and enhanced cardiac evaluation even among asymptomatic patients that may detect cardiac sequelae and guides prognostication and therapies in this overlooked disease.

## Supporting information

**S1 Table. American Heart Association Classification of Chagas Cardiomyopathy.** Arrhythmias and conduction disease can occur from B1 through D stages. HF, heart failure; NYHA, New York Heart Association. Adapted from Nunes et al [2].
(DOCX)

**S2 Table. Rassi Risk Score to Predict Mortality Related to Chagas Disease.** *NYHA, New York Heart Association. Adapted from Keegan et al [36].
(DOCX)

**S3 Table. Proportion of ECG Abnormalities by Age Group.**
(DOCX)

**S4 Table. American Heart Association Stage of Cardiomyopathy by Age Group.**
(DOCX)

## Author Contributions

**Conceptualization:** Katherine A. Reifler, Natasha S. Hochberg, Davidson H. Hamer.

**Data curation:** Katherine A. Reifler, Samantha M. Hall, Shahzad Hassan, John A. Bostrom.

**Formal analysis:** Katherine A. Reifler, Samantha M. Hall.

**Methodology:** Katherine A. Reifler.

**Project administration:** Alejandra Salazar.

**Supervision:** Malwina Carrion, Natasha S. Hochberg, Davidson H. Hamer, Deepa M. Gopal, Daniel Bourque.

**Writing – original draft:** Katherine A. Reifler.

**Writing – review & editing:** Katherine A. Reifler, Alyse Wheelock, Samantha M. Hall, Elizabeth D. Barnett, Natasha S. Hochberg, Davidson H. Hamer, Deepa M. Gopal, Daniel Bourque.

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
