## [Decision Letter · Decision Letter 0]

25 Oct 2023

Dear Dr. Reifler,

Thank you very much for submitting your manuscript "Chagas Cardiomyopathy in Boston, Massachusetts: identifying disease and improving management after community and hospital-based screening" for consideration at PLOS Neglected Tropical Diseases. As with all papers reviewed by the journal, your manuscript was reviewed by members of the editorial board and by several independent reviewers. In light of the reviews (below this email), we would like to invite the resubmission of a significantly-revised version that takes into account the reviewers' comments. 

We cannot make any decision about publication until we have seen the revised manuscript and your response to the reviewers' comments. Your revised manuscript is also likely to be sent to reviewers for further evaluation.

Sincerely,

Pierre Buekens

Academic Editor

Charles Jaffe

Section Editor

Thank you for submitting this interesting manuscript. Please respond to the reviewers comments and suggestions.

Reviewer's Responses to Questions

**Key Review Criteria Required for Acceptance?**

**Methods**

-Are the objectives of the study clearly articulated with a clear testable hypothesis stated?

-Is the study design appropriate to address the stated objectives?

-Is the population clearly described and appropriate for the hypothesis being tested?

-Is the sample size sufficient to ensure adequate power to address the hypothesis being tested?

-Were correct statistical analysis used to support conclusions?

-Are there concerns about ethical or regulatory requirements being met?

Reviewer #1: This is a single centre retrospctive cohort analysis of 109 patients with a diagnosis of Chagas disease at different stages. This is a descriptive observational study with no real hypothesis. The main objective was to describe the rate of cardiac complications reportedly related with Chagas disease in a cohort of Central American migrants living in a non-endemic region.

Reviewer #2: Classification: I'm not sure if the classification of asymptomatic equates with indeterminate Chagas, could the authors explain more? It seems that this was patient-reported although other possibilities (e.g. LVEF, ECG alterations) seem to be available. Had authors considered dividing by a different categorization, e.g. anyone with abnormal ECG findings or use AHA categories. Also, sometimes it was hard for the reader to orient which category was being described or analyzed at a given time.

Data analysis: The sample size seems large enough to support simple comparisons of proportions (chi squared), or Fisher's exact could be used for small cells. The value of this would depend a great deal on the categories above. For example, to see differences between indeterminate and patients with progression. (Or symptomatic vs. asymptomatic).

**Results**

-Does the analysis presented match the analysis plan?

-Are the results clearly and completely presented?

-Are the figures (Tables, Images) of sufficient quality for clarity?

Reviewer #1: As this is a retrospective cohort study no pre-planned statistical analysis was performed. Findings are clearly reported, only descriptive as the sample is too small to perform any formal statistical analysis. Tables and figures are appropriate.

Reviewer #2: The results are comprehensive -- sometimes it was hard to oriented toward which category was being described.

**Conclusions**

-Are the conclusions supported by the data presented?

-Are the limitations of analysis clearly described?

-Do the authors discuss how these data can be helpful to advance our understanding of the topic under study?

-Is public health relevance addressed?

Reviewer #1: The authors concluded that in this retrospective series of individuals with serologic evidence of t. cruzi infection of a population composed of mostly Central American migrants, the incidence of cardiac complications was high in a non-endemic region. The authors suggest that Chagas disease diagnosis

prompts cardiac evaluation which often identifies actionable cardiac disease and provides opportunities for prevention and treatment.The discussion is lengthy and addresses the need for a systematic assessment of cardiac function in patients with as serologic diagnosis of Chagas disease regardless of the presence of symptoms.

Reviewer #2: Yes, but the authors could comment more on treatment and care options for indeterminate patients.

**Editorial and Data Presentation Modifications?**

Reviewer #1: Minor issues:

Line 199; In the patients with stroke under the age of 40 was the TTE normal?

Table 3. How many of the patients with Stroke had AF detected before and after the stroke? Were these all ischemic strokes? If so state this.

Line 238; Please provide further information of the ambulatory monitoring including duration of monitoring. Was the PVC burden calculated if so report this. When you indicate atrial arrhythmias what are you reporting? AF, Atrial flutter, atrial tachycardia? What do you mean by skipped neats, this is not a cardiology term, premature atrial complexes extrasystoles atrial or ventricular? Please clarify. Pause are defined how > 3s symptomatic, symptomatic > 6 s? Diurnal or nocturnal during sleep?

Line 249 ETT; Did patients with chronotropic incompetence have any symptoms? Please report if this is the case. Similarly, patients with Exercise induced ischemia had any demonstrated obstructive CAD?

Line 259 CMRI: Only 12 patients had CMRI, it is unclear what were the indications and why so many pf the symptomatic patients did not have this test, please elaborate. Similarly, please report the burden of LGE as you do mention the finding of LGE and fibrosis.

Line 271: Cardiac Catheterisation: Only 10 symptomatic patients underwent catheterisation and only 30% had some obstructive CAD, no mention on treatment for these patients. Similarly, patients with no evidence of obstructive CAD "suggested" that Chagas was the cause. Please elaborate on how you came to this conclusion, did you evidence vasospasm, is this due to distal microcirculation disease? Any insights on the characteristics of coronary flow in these patients? i.e. slow flow ecstatic coronaries etc which have been reported previously in Chagas cardiomyopathy patients?

Line 282: 2 patients with ICD were admitted for ablation and cardio version. I assume cardio version was for AF as this is not treated by the ICD, ablation of VT? What other treatment strategies were tried prior to ablation? i.e. amiodarone or other anti arrhythmic therapy? As this is a merely descriptive report these details are essential.

Discussion

Line 346 ref 23 is a bit outdated several more recent studies have reported the Global Longitudinal strain detected by TTE has a significant predictive value for detecting early cardiomyopathy in patients with Chagas initially classified in Stage A. Echeverría LE, Rojas LZ, Villamizar MC, et al Echocardiographic parameters, speckle tracking, and brain natriuretic peptide levels as indicators of progression of indeterminate stage to Chagas cardiomyopathy. Echocardiography. 2020 Mar;37(3):429-438. doi: 10.1111/echo.14603. Please review and update references. Similarly, the BENEFIT trial published a Echo subsidy that identified a simple and reproducible Wall Motion's Score that predicted mortality and cardiovascular events. Please review and include this reference. Schmidt A, Dias Romano MM, Marin-Neto JA, et al. Effects of Trypanocidal Treatment on Echocardiographic Parameters in Chagas Cardiomyopathy and Prognostic Value of Wall Motion Score Index: A BENEFIT Trial Echocardiographic Substudy. J Am Soc Echocardiogr. 2019 Feb;32(2):286-295.e3. doi: 10.1016/j.echo.2018.09.006.

Reviewer #2: Abstract: In Results, it would be good to have absolute numbers for the numerators and denominators along with the percentages. (For example do the percentages of symptoms reflect the entire sample or just the 21% who were diagnosed due to symptoms and ECG abnormalities?)

63-64: Maybe this statement should be contextualized more, since there are PAHO and AHA recommendations and consensus guidelines in Brazil and elsewhere. Guidelines for the Diagnosis and Treatment of Chagas Diseases (paho.org), 2 nd Brazilian Consensus on Chagas Disease, 2015 - PubMed (nih.gov), Chagas Cardiomyopathy: An Update of Current Clinical Knowledge and Management: A Scientific Statement From the American Heart Association | Circulation (ahajournals.org)

80-81: We don;t really have evidence that it’s growing in the 2020’s (and reference 3 wouldn’t really support that statement), although we do have an estimate of the burden in people of Latin American origin from the US as mentioned already – I would just leave it at that, and mention that the US also has limited, occasional evidence of both congenital and autochthonous vector transmission. 

93-95 and is also not recommended in current PAHO recommendations

126 – could include what test was used at CDC (Wiener rec. v. 3?, and any other test to confirm positives?)

157 “was” classified

156-159: definition of symptomatic was whether symptoms suggestive of Chagas disease were reported by the patient? Or this was based on review of patients’ medical records? For Table 1, I wonder if you considered dividing by a different categorization, e.g. anyone with abnormal ECG findings or use AHA categories. 

Figure 1: the text and caption indicate 25, not 52? Some patients have multiple fundings?

345-347: Interesting finding, how does it compare to what is in the literature?

386: does this 51% reflect the “asymptomatic” group from Table 1? 

Discussion: maybe you could also comment more on why it is important to identify patients in the indeterminate stage, and what options for treatment are available.

**Summary and General Comments**

Reviewer #1: This is a retrospective cohort study that included 109 patients with Chagas diseasee at different stages. Strengths include that for the US this is a large sample and highlights the fact that almost half of the patients ended with some form of cardiomyopathies alteration. This is worth highlighting. On the other hand the study has several limitations with many patients having very few diagnostic tests which limits the use of this data to provide any sort of recommendation on which should be the appropriate diagnostic approach on this often selected population. The authors should perform a more systematic approach and propose an approachable diagnostic algorithm based on their findings. Both the role of GLS and calculating the Wall Motion Index score suggested above should be performed and the manuscript adjusted accordingly.

Reviewer #2: This is an interesting, well written and important study following a cohort of patients with Chagas disease in Boston. The study makes an important contribution to the literature on Chagas disease in the US. The different classifications used were sometimes hard to follow; the paper could be strengthened by choosing one classification based on the literature (e.g. AHA stages?) and just ensuring the structure of the paper facilitates clarity about which group is being described where.

PLOS authors have the option to publish the peer review history of their article (what does this mean?). If published, this will include your full peer review and any attached files.

Reviewer #1: No

Reviewer #2: No
---

## [Decision Letter · Decision Letter 1]

11 Jan 2024

Dear Dr. Reifler,

We are pleased to inform you that your manuscript 'Chagas cardiomyopathy in Boston, Massachusetts: identifying disease and improving management after community and hospital-based screening' has been provisionally accepted for publication in PLOS Neglected Tropical Diseases.

Best regards,

Pierre Buekens

Academic Editor

Charles Jaffe

Section Editor

Thank you for responding adequately to all comments and suggestions.

Reviewer's Responses to Questions

**Key Review Criteria Required for Acceptance?**

**Methods**

-Are the objectives of the study clearly articulated with a clear testable hypothesis stated?

-Is the study design appropriate to address the stated objectives?

-Is the population clearly described and appropriate for the hypothesis being tested?

-Is the sample size sufficient to ensure adequate power to address the hypothesis being tested?

-Were correct statistical analysis used to support conclusions?

-Are there concerns about ethical or regulatory requirements being met?

Reviewer #1: This is a revised manuscript of a retrospective cohort study that included 109 patients with Chagas disease at different stages. Strengths include that for the US this is a large sample and highlights the fact that almost half of the patients ended with some form of cardiomyopathic alteration. The authors have provided a detailed response to all the previous reviewers comments and have markedly improved the manuscript.

Study design is appropriate and objectives were clearly described. Population included was clearly described. I have no ethical concerns.

Reviewer #2: (No Response)

**Results**

-Does the analysis presented match the analysis plan?

-Are the results clearly and completely presented?

-Are the figures (Tables, Images) of sufficient quality for clarity?

Reviewer #1: This study provided descriptive analysis that is appropriate for the type of study.

Results have been revised and are clearly stated. Figures and tables are clear.

Reviewer #2: (No Response)

**Conclusions**

-Are the conclusions supported by the data presented?

-Are the limitations of analysis clearly described?

-Do the authors discuss how these data can be helpful to advance our understanding of the topic under study?

-Is public health relevance addressed?

Reviewer #1: Conclusions are supported by the data presented.

Reviewer #2: (No Response)

**Editorial and Data Presentation Modifications?**

Reviewer #1: None this revised manuscript appropriately addresses all the prior concerns raised by the reviewers.

Reviewer #2: (No Response)

**Summary and General Comments**

Reviewer #1: This is a well-written manuscript that addresses a topic mostly neglected in the US and provides insight into the significant complications in patients ion different stages of Chagas disease. this is an important manuscript and merits publication in the current state.

Reviewer #2: The authors addressed all my previous comments appropriately, and the revised version is much clearer and easier to follow.

PLOS authors have the option to publish the peer review history of their article (what does this mean?). If published, this will include your full peer review and any attached files.

Reviewer #1: No

Reviewer #2: No

---

## [Editor Report · Acceptance letter]

17 Jan 2024

Dear Dr. Reifler,

We are delighted to inform you that your manuscript, "Chagas cardiomyopathy in Boston, Massachusetts: identifying disease and improving management after community and hospital-based screening," has been formally accepted for publication in PLOS Neglected Tropical Diseases.

Best regards,

Shaden Kamhawi

co-Editor-in-Chief

Paul Brindley

co-Editor-in-Chief
